# Treatment with Argovit^®^ Silver Nanoparticles Induces Differentiated Postharvest Biosynthesis of Compounds with Pharmaceutical Interest in Carrot (*Daucus carota* L.)

**DOI:** 10.3390/nano11113148

**Published:** 2021-11-22

**Authors:** Laura Sofia Santoscoy-Berber, Marilena Antunes-Ricardo, Melissa Zulahi Gallegos-Granados, Juan Carlos García-Ramos, Alexey Pestryakov, Yanis Toledano-Magaña, Nina Bogdanchikova, Rocio Alejandra Chavez-Santoscoy

**Affiliations:** 1Tecnológico de Monterrey, Escuela de Ingeniería y Ciencias, Av. Eugenio Garza Sada 2501 Sur, Monterrey 64849, Mexico; A01630849@itesm.mx (L.S.S.-B.); marilena.antunes@tec.mx (M.A.-R.); 2Facultad de Ciencias Químicas e Ingeniería, Universidad Autónoma de Baja California (UABC)—Campus Tijuana, Calzada Universidad 14418, Parque Industrial Internacional Tijuana, Tijuana 22390, Mexico; zulahi.gallegos@uabc.edu.mx; 3Escuela de Ciencias de la Salud, UABC, Blvd. Zertuche y Blvd., De los Lagos S/N Fracc, Valle Dorado, Ensenada 22890, Mexico; juan.carlos.garcia.ramos@uabc.edu.mx (J.C.G.-R.); yanis.toledano@uabc.edu.mx (Y.T.-M.); 4Research School of Chemistry and Applied Biomedical Sciences, Tomsk Polytechnic University, 634050 Tomsk, Russia; pestryakov2005@yandex.ru; 5Centro de Nanociencias y Nanotecnología, Universidad Nacional Autónoma de México (UNAM), Carretera Tijuana-Ensenada Km 107, Ensenada 22860, Mexico; nina@cnyn.unam.mx

**Keywords:** silver nanoparticles, postharvest abiotic stress, phenolic compounds, *Daucus carota*

## Abstract

The global market for plant-derived bioactive compounds is growing significantly. The use of plant secondary metabolites has been reported to be used for the prevention of chronic diseases. Silver nanoparticles were used to analyze the content of enhancement phenolic compounds in carrots. Carrot samples were immersed in different concentrations (0, 5, 10, 20, or 40 mg/L) of each of five types of silver nanoparticles (AgNPs) for 3 min. Spectrophotometric methods measured the total phenolic compounds and the antioxidant capacity. The individual phenolic compounds were quantified by High Performance Liquid Chromatography (HPLC) and identified by –mass spectrometry (HPLC-MS). The five types of AgNPs could significantly increase the antioxidant capacity of carrots’ tissue in a dose-dependent manner. An amount of 20 mg/L of type 2 and 5 silver nanoparticle formulations increased the antioxidant capacity 3.3-fold and 4.1-fold, respectively. The phenolic compounds that significantly increased their content after the AgNP treatment were chlorogenic acid, 3-*O*-caffeoylquinic acid, and 5′-caffeoylquinic acid. The increment of each compound depended on the dose and the type of the used AgNPs. The exogenous application of Argovit^®^ AgNPs works like controlled abiotic stress and produces high-value secondary bioactive compounds in carrot.

## 1. Introduction

Health agencies worldwide have encouraged the population to consume vegetables and fruits as part of a healthy diet. Many reports have described that the sufficient daily consumption of vegetables and fruits could help prevent chronic diseases: heart disease, cancer, diabetes, obesity, several micronutrient deficiencies, etc. [1]. In addition, the reports suggest that the prevention of chronic diseases by consuming vegetables and fruits is related to their content of secondary metabolites, to which a beneficial bioactivity for health is attributed [2,3,4]. For example, phenolic compounds have been widely reported for their antioxidant, antiproliferative, and hepatoprotective activity, among others [5,6,7]. Due to their properties, these secondary metabolites represent an important impact on the global economy. According to a British Broadcasting Corporation (BBC) report, the global market for plant-derived bioactive compounds will grow with an annual growth rate of 6.1% between 2017 and 2022 [8].

The use of controlled abiotic stress to activate the secondary plant metabolism and enhance the nutraceutical content have been previously reported [9]. Abiotic stress produces reactive oxygen species (ROS) in plant cells, whereby the adaptative response of plant cells maintains homeostasis in plants. Increasing the ROS levels in the plant cell (>10 nM) due to postharvest abiotic stress breaks the homeostasis and generates hormesis, thereby enhancing the phytochemical content in the crop. At the same time, product quality could be maintained, enhanced, or decreased [9,10,11,12]. The most common technologies to increase the phytochemical content of crops are modified atmospheres, phytohormones, and ultraviolet (UV) radiation [13,14]. However, these technologies do not necessarily offer differential effects in the production of nutraceutical compounds.

Carrot is a horticultural crop characterized by increasing the content of phenolic compounds when subjected to postharvest abiotic stress [13]. Moreover, the health benefits of phenolic compounds found in horticultural crops, particularly in carrots, have encouraged the food industry to incorporate them into processed foods to enhance their nutraceutical quality. Finding innovative methods that allow the differentiated production of bioactive compounds at reasonable costs would permit the food industry to focus functional products on specific niche markets with particular health needs, and thus contribute to personalized nutrition.

Nanoparticles (NPs) (size between 1 and 100 nm), at very low doses, have the potential to boost the plant metabolism [15]. It has been reported that NPs positively impact plant metabolism, leading to the increase in total biomass, yield, and hence the rate of harvested high-value bioactive compounds in vivo and in vitro [16]. Silver nanoparticles (AgNPs) have been reported to play an essential role in enhancing seed germination and plant growth [12,17,18]. Moreover, it has been reported that the exogenous application of silver nanoparticles increased the content of antioxidant compounds and fatty acids on sunflower [19]. Other nanomaterials have also been used to improve the content of secondary metabolites. For example, metal oxide nanoparticles (NiO, CuO, and ZnO) have been reported to increase the anthocyanin, the total phenolic, and the total flavonoid contents on *Brassica rapa* ssp. [20].

Until now, only some nanoelicitors have been subjected to studies in various in vitro and in vivo systems to determine the possible risks in humans and the environment. Our research group has been working with AgNP formulations stabilized with polyvinylpyrrolidone (PVP) as coating agent. These AgNPs possess a defined (Ag)/(coating agent ratio) that contributes to their low toxicity on reference systems like *Allium cepa* [21], human peripheral blood erythrocytes [22] and lymphocytes [23], and mice [24,25]. The oral lethal dose study shows that the five AgNP formulations investigated in the present study are included in Category 4 (>300 to ≤ 2000 mg/Kg bodyweight) or Category 5 (>2000 to ≤ 5000 mg/Kg bodyweight) of the Globally Harmonized System of Classification and Labelling of Chemicals [24].

It has not been previously reported how the differences in the size of nanoparticles with the same composition are capable of differently influencing the production of specific phenolic compounds with pharmaceutical value. Therefore, one objective of this study was to analyze the *Daucus carota* L. differentiated response in phenolic compound production and the antioxidant activity elicited by the exposure to four doses of five types of fully characterized PVP-AgNP formulations that possess a defined (Ag)/(coating agent ratio).

## 2. Materials and Methods

### 2.1. Materials

The five Argovit^®^ AgNPs used in this study are polyvinylpyrrolidone (PVP)-AgNPs formulations in stable aqueous suspensions with an overall stock concentration of 200 mg/mL (metallic silver + PVP) in distilled water. For this study, the formulations were labeled as AG1, AG2, AG3, AG4, and AG5. According to the manufacturer, differences between each AgNP formulation consist of the molecular mass of PVP used as coating agent or the synthesis conditions as follows. AG1: PVP K-15; AG2: PVP K-17; AG3: PVP K-17 with higher radiation potency used for the synthesis; AG4: PVP K-30; AG5: PVP 12.6 ± 2.7 KDa. Molecular masses of PVP K-15, K-17, and K-30 PVP (Boai NKY Pharmaceuticals Ltd., Jiaozuo, China) are 8000–12,000 kDa, 10,000–16,000 kDa, and 45,000–58,000 kDa, respectively. The morphology, zeta potential, average diameter, and thermogravimetric analysis of used silver nanoparticles have been previously reported [24]. 

Briefly, AgNPs composition determined by TGA analysis shows silver content of 1.14–1.32%, PVP: 19.6–24.5%, and H_2_O 74.2–79.2% *w*/*w*. All formulations present a spherical morphology with a distribution size of nanoparticles within the range of 5–80 nm. The average size of each formulation determined by Transmission Electron Microscopy (TEM) (JEM-2010,JEOL©, CDMX, Mexico) was AG1 = 16.4 ± 8.0, AG2 = 25.4 ± 13.2, AG3 = 19.0 ± 9.3, AG4 = 16.4 ± 8.1, and AG5 = 30.6 ± 23.2. Zeta potential of the fAg1-Ag5 formulations was within the range −0.46 to −5.13 mV [24].

### 2.2. Plant Material and Processing

Carrots (*D. carota*) were obtained from a local market (CALIMAX, Tijuana, Mexico), sorted, washed, and disinfected with chlorinated water (250 ppm, pH 6.5). 

The whole carrot samples were immersed in different concentrations (0, 5, 10, 20, or 40 mg/L) of each of five types of silver nanoparticle for 3 min, then the samples were left to rest for 24 h out of the solution at room temperature, and finally, the preparation for the phytochemical analysis was carried out. The negative control treatment consisted of carrot samples immersed in water (0 mg/L of silver nanoparticles). These experiments were performed with at least 10 replicates. From here on, the concentration units for AgNP formulations consider the metallic silver content on each formulation, unless it is mentioned the contrary.

### 2.3. Sample Preparation for Phytochemical Analyses

Five grams of carrot tissue was homogenized with 20 mL methanol using a homogenizer (VWR^®^ 200 Homogenizer, CDMX, Mexico) and centrifuged at 29,000× *g* for 15 min at 4 °C. The obtained methanolic extract was used to analyze total soluble phenolics and antioxidant activity (ORAC value). To identify and quantify the individual phenolic compounds, the methanolic extracts were filtrated through nylon membranes (0.2 μm) prior to injection to the chromatographic systems.

### 2.4. Total Phenolic Content and Determination of Antioxidant Capacity

Total phenolic content was determined as reported previously [26]. Briefly, methanolic extracts were diluted with distilled water in a 96-well microplate, followed by the addition of 0.25 N Folin–Ciocalteu reagent (Sigma-Aldrich, Saint Louis, MO, USA). The mixture was left for 3 min, and then 1 N Na_2_CO_3_ was added. The final mixture was incubated for 2 h at room temperature under dark conditions. Spectrophotometric readings at 725 nm were collected using a plate reader (SPECTROstar Omega, BMG Labtec, Cary, NC, USA). Total phenolics were expressed as µg chlorogenic acid equivalents/g of fresh tissue. The antioxidant activity was determined with the oxygen radical absorbance capacity (ORAC) assay (Sigma-Aldrich, Saint Louis, MO, USA). The ORAC value was obtained by using the procedure reported before [13]. All data were expressed as micromoles of Trolox equivalents per gram of fresh tissue (µmol of TE/g). 

### 2.5. Quantification of Individual Phenolic Compounds by HPLC

Chromatographic analyses of samples were performed in an Agilent HPLC system (Agilent Technologies, Santa Clara, CA, USA) with a reverse phase C18 column (Zorbax Eclipse XDB-C18, 150 mm × 4.6 mm i.d., 5 μm) maintained at 25 °C. Elution solvent phases consisted of A: 0.1% formic acid (*v*/*v*) in water and B: methanol. Separation was achieved using an initial solvent composition of 35% (B) during 5 min, increased to 60% (B) within 15 min, and subsequently ramped to 90% (B) within 30 min, decreased to 35% (B) in 5 min to re-equilibration. The flow rate was established at 0.8 mL/min and the injection volume was 20 μL. The UV-vis photodiode array detector was set at 280 nm, 320 nm, and 365 nm. Standard solutions of gallic acid (y = 112.2x − 4325.2 R^2^ = 0.9993), chlorogenic acid (y = 31.643x – 149.3 R^2^ = 0.9999), and quercetin (y = 65.5x + 68.738 R^2^ = 0.9979) were used for the quantification of phenolic compounds detected at 280 nm, 320 nm, and 365 nm, respectively. Since the most abundant phenolic compounds were detected at 320 nm, quantification was expressed as equivalents of chlorogenic acid (CA). The selectivity of phenolic compounds production was calculated by the contribution (percentage) of the concentration of each compound to the sum of concentrations of all identified phenolic compounds.

### 2.6. Analysis of Individual Phenolic Compounds by HPLC-MS

Individual phenolic compounds were separated on a Luna 5u C18 column of 150 mm × 4.60 mm (Phenomenex, Agilent, Santa Clara, CA, USA) by an HPLC coupled with a diode array detector (DAD) (Agilent, Santa Clara, CA, USA). The chromatograms were obtained at 280 nm, 320 nm, and 365 nm, respectively. The mobile phase was A: acidic water (0.1% phosphoric acid) and B: acetonitrile-phosphoric acid (0.1% phosphoric acid). The flow rate was 1.0 mL/min at 40 °C. The gradient elution was 5% B at 0 min and 50% B at 30 min. Quantification of each compound was carried out with ferulic acid equivalents.

The identification of phenolic acids was confirmed using an HPLC coupled with time-of-flight mass spectrometry (LC/MS-TOF) (Agilent Technologies, Santa Clara, CA, USA) equipped with a negative electrospray ionization source. For the identification of compounds, the Dictionary of Natural Products and Mass Bank databases were consulted.

### 2.7. Quantification of Silver Content in Carrot Tissue after Exposure by ICP-OES

At the end of each period of exposition, carrots were washed with 20 mM EDTA-Na2 solution to remove the silver nanoparticles that were on the surface, and then were rinsed with distilled water. Five grams of tissue of carrot per treatment were measured with an electronic balance (VELAB VE-204, CDMX, Mexico) with an accuracy of 0.01 g. Then, the tissue samples were dried in an oven with air flow for 72 h at 60 °C, followed by 1 day at 70 °C. Subsequently, the tissue of each treatment (500 mg) was digested with 10 mL of nitric acid (85% *v*/*v*) overnight. Resulting digests were diluted up to 10 mL with deionized water and then metallic silver concentrations were determined by an inductively coupled plasma optical emission spectrophotometer (ICP-OES 400, Perkin-Elmer, Richmond, CA, USA). The detection limit in ICP was of 0.22 mg/L. All samples were analyzed at λ = 328.068 nm. Each sample was run in triplicate to guarantee that the measured absorbencies were constant. Metal concentrations, calculated from each replicate absorbance value, were then used to calculate an average metal sample concentration. The concentration of metallic silver in plant tissues is expressed in μg/g on a dry weight (dw) basis.

### 2.8. Statistical Analysis

GraphPad Prism version 9.00 (GraphPad Software, San Diego, CA, USA, 5 November 2020) was used to analyze data, expressed as the means ± standard error. One-way ANOVA statistical analysis was performed. Significant differences were considered with *p* < 0.05. A Tukey’s test was made to identify significant differences among groups.

## 3. Results

### 3.1. Total Phenolic Content and Antioxidant Capacity

Figure 1 presents the antioxidant capacity of the carrot tissue stimulated with the five AgNP formulations measured by the ORAC technique. Carrots were exposed to four different concentrations of the corresponding AgNPs formulation for three minutes compared with the negative control treatment. The negative control treatment consisted of carrot samples immersed in water (0 mg/L of silver nanoparticles).

AgNPs Argovit^®^ were able to increase the antioxidant capacity in a dose-dependent manner. The hormetic effect of AgNPs Argovit^®^ was observed, since the antioxidant capacity increased with increasing doses, and after a specific dose, the antioxidant capacity significantly decreased when the concentration of nanoparticles increased. Each AgNP stimulated the carrot tissue differently, increasing its antioxidant capacity in different doses. AG1 increased the antioxidant capacity up to 2-fold compared with the control treatment at a dose of 10 mg/L. AG2 showed the maximum antioxidant capacity at an amount of 20 mg/L and was 2.8-fold greater than the control treatment. AG3 and AG4 increased antioxidant capacity up to 1.5–1.8-fold at a dose of 20 mg/L and 10 mg/L, respectively. AG5 was the most effective treatment in improving the antioxidant capacity. It managed to increase the antioxidant capacity up to 3.25-fold at a dose of 20 mg/L.

Additionally, in Figure 2 is presented the total phenolic content measured in carrot samples by the Folin-Ciocalteu method in carrot tissue. The stimulus was a 3-min immersion in five types of AgNPs at four doses and the negative control treatment.

The hormetic effect was also observed in the phenolic content of the carrot tissue treated with AgNPs. However, the behavior of phenolic compounds differs from that of the antioxidant capacity. AG2, AG3, and AG5 increased the total phenolic content by 2.4-fold at a concentration of 20 mg/L. In contrast, AG1 increased the total phenolic content by 1.4-fold and AG3 by 2-fold, at a dose of 10 mg/L and 20 mg/L, respectively. The differences observed in the increments of the phenolic compounds compared to the increments in the antioxidant capacity suggested that the specific phenolic compounds obtained in each treatment could be differentiated. That is the reason the phenolic compounds were measured individually by an HPLC.

### 3.2. Identification of Phenolic Compounds in Carrots 

Figure 3 shows the typical HPLC chromatography obtained for the samples to identify phenolic compounds (280 nm) in carrots used in the present study. Each compound was identified by comparing the m/z+ of fragments obtained in the mass spectrometry with the reported mass spectral characteristics of the phenolic compound reported for carrots (Table 1).

### 3.3. Quantification of Phenolic Compounds in Carrots and Selectivity 

Figure 4 presented the phenolic compounds determined by an HPLC coupled to a UV-vis detector identified in each stimulated carrot tissue with the AgNP formulation at the four tested concentrations and the negative control treatment, which consisted of carrot samples immersed in water (0 mg/L of silver nanoparticles). The methanolic extract probably contains not only phenolic compounds; however, in the present study the aim was to identify phenolic compounds, especially ones with pharmaceutical interest.

Figure 5 shows the selectivity to induce the content of chlorogenic acid, 3-*O*-caffeoylquinic acid, 5′-caffeoylquinic acid, and ferulic acid. These compounds were the compounds with the highest selectivity. The chlorogenic acid reached the highest selectivity (39.36%) with the AG4 treatment at concentrations of 40 mg/L. In addition, chlorogenic acid achieved a significant selectivity with AG3 at concentrations of 5 and 10 mg/L (32.31% and 36.16%, respectively), and with AG4 at a dose of 20 mg/L (38.47%). The 3-*O*-caffeoylquinic acid obtained the highest selectivity of 37.92% stimulated with AG1 at a concentration of 5 mg/L. For 5′-caffeoylquinic acid, the highest selectivity (33.82%) was obtained with the AG3 treatment at a dose of 20 mg/L. The ferulic acid selectivity only increased with the AG2 stimulus at a dose of 10 mg/L (12.00% selectivity) and AG4 at a concentration of 40 mg/L with a selectivity of 11.80%.

### 3.4. Quantification of Silver after Carrots Exposure to AgNPs Formulations

The quantification of silver by ICP-OES was performed in carrots exposed to the highest concentration used in this study (40 mg/L) and was compared with the control group. The results show that the silver content of the control and the treated groups were always below the detection level.

## 4. Discussion

The five types of AgNPs could significantly increase the antioxidant capacity of carrots’ tissue (Figure 1). The increase of antioxidant capacity is dose-dependent. Previously, reports have described that silver nanoparticles stabilized with PVP could significantly increase the antioxidant capacity [12,23]. The treatments AG2 and AG5 reached the highest antioxidant capacity in carrot tissue. Treatments with 20 mg/L of AG2 and AG5 increased the antioxidant capacity 3.3-fold and 4.1-fold, respectively (Figure 1). 

The phenolic content of carrot also increased with the AgNP concentration growth for the five types of AgNPs (Figure 2). However, the increased phenolic content (Figure 2) did not correlate with the increased antioxidant capacity (Figure 1). This is attributed to the fact that the profile of the compounds produced in each treatment and each concentration were different (Figure 4). It is essential to mention that the antioxidant response of carrots was observed despite the short exposure time assessed. The absence of silver quantified by ICP-OES in carrot tissues exposed to the highest concentration evaluated in this work suggests that the effect elicited by the AgNP formulations is swift and does not imply a significant silver incorporation into the carrot tissue. These factors could be extremely helpful in the design of noninvasive strategies to produce valuable carrot secondary metabolites.

The compounds that increased their contents to the greatest extent after AgNP treatments were chlorogenic acid, 3-*O*-caffeoylquinic acid, and 5′-caffeoylquinic acid. Chlorogenic acid has been studied for many potential health benefits, including its antidiabetic, antiproliferative, anti-inflammatory, and antiobesity effects [27]. 3-*O*-caffeoylquinic acid is an antioxidant marker in several plants [28] that have been reported as an essential precursor of flavonoids with anti-inflammatory effects [29]. Additionally, 3-*O*-caffeoylquinic acid has been known for its regulatory effects in lipid metabolism [30]. 5′-Caffeoylquinic acid has been also described as a precursor of important flavonoids such as quercetin with antioxidant, anticarcinogenic, and anti-inflammatory effects [31,32,33]. Meanwhile, ferulic acid has been widely reported for its antioxidant, anticarcinogenic, neuroprotective, antidiabetic, hepatoprotective, and cardioprotective effects [34,35,36]. The overproduction of three out of four of these bioactive compounds in carrots after the treatment with AgNPs represents a nonpharmacological and noninvasive approach for treating or preventing some chronic diseases. Other reports have described the increasing of lipidic secondary metabolism due to stimuli with silver nanoparticles at different doses on sunflower. Authors reported that 60 mg/L of the complete formulation concentration of the tested green-synthesized AgNPs improved the biochemical, fatty acid, and enzymatic attributes of sunflower plants [19].

Nanomaterials, particularly Argovit^®^ silver nanoparticles, can change the agronomic characteristics of plants, such as plant growth, biomass, and shoot and root length, among others [10,18,23]. These physiological parameters directly influence the yield and quality obtained from a crop. Additionally, silver nanoparticles, such as the Argovit^®^ nanoparticles, could be used as a source of controlled abiotic stress to increase the directed bio-production of secondary metabolites of pharmaceutical and nutritional interest.

Exogenous applications of Argovit^®^ silver nanoparticles work like controlled abiotic stress and produce reactive oxygen species (ROS) in carrots. Then, the adaptive response of plant cells leads to maintaining homeostasis in plants. However, the increasing ROS levels break the homeostasis and generates hormesis, resulting in the enhancement of the phytochemical content in the carrot. The hormetic response, due to other controlled abiotic stresses of carrots and other crops, has been widely reported before. Regarding the Argovit^®^ silver nanoparticles, the dose-dependent effect and generation of hormesis was suggested already for vanilla [10] and sugarcane [12]. In addition, a dose-dependent effect in the reactive oxygen species (ROS) generation, malondialdehyde production, and anthocyanin biosynthesis due to AgNPs in *Brassica rapa* seedlings has been also reported. The results suggest that authors also observed a hormetic effect in their experiments [20].

Under extreme stress conditions, horticultural crops can act as biofactories of nutraceuticals, which can be extracted and commercialized in dietary supplements or food industries. The present work described how nanomaterials, particularly Argovit^®^ silver nanoparticles AG2 and AG5, could be used as abiotic elicitors to improve the yields of plant-derived bioactive compounds of great pharmaceutical and/or nutritional interest.

## 5. Conclusions

The Argovit^®^ silver nanoparticles studied in the present work were able to induce the production of secondary metabolites in a differentiated way in carrots. The difference in the production of metabolites was due to the small changes in the physicochemical properties of the Argovit^®^ silver nanoparticles (AG1, AG2, AG3, AG4, AG5) and the dose of each nanoparticle applied endogenously.

The treatments with the silver nanoparticles AG2 and AG5 led to the highest increase (3.3-fold and 4.1-fold, respectively) of antioxidant capacity and bioactive compounds, such as chlorogenic acid, 3-*O*-caffeoylquinic acid, 5′-caffeoylquinic acid, and ferulic acid. These acids exhibit different biological activities such as anti-inflammatory, anticarcinogenic, antidiabetic, hepatoprotective, cardioprotective, and neuroprotective actions, etc. Therefore, using these nanoparticles for carrot postharvest treatment is an effective approach for overproducing these bioactive compounds and furthering their use in pharmaceutical and/or nutritional areas.

Unlike most AgNP formulations, the Agrovit^®^ formulation demonstrated low genotoxicity and cytotoxicity in sensitive plants such as *Allium cepa* [21]. It has also been reported that the Agrovit^®^ nanoparticles improve the physiological characteristics of plants such as vanilla and sugar cane obtained by micropropagation [10,12]. In the present study, these formulations demonstrated a significant effect in the overproduction of bioactive compounds in a postharvest crop such as *Daucus carota*. The effect elicited by the AgNP formulations is swift and requires minimal concentrations and exposure time. These factors could be extremely helpful in the design of nontoxic and noninvasive strategies to produce valuable plant secondary metabolites. Thus, Agrovit^®^ can be exogenously applied in postharvest crops to increase the performance of bioactive compounds with pharmaceutical interest. Nevertheless, a deep understanding of the role of AgNPs in plant physiology at the molecular level is still lacking, and further studies should generate more information about it.

## Figures and Tables

**Figure 1 nanomaterials-11-03148-f001:**
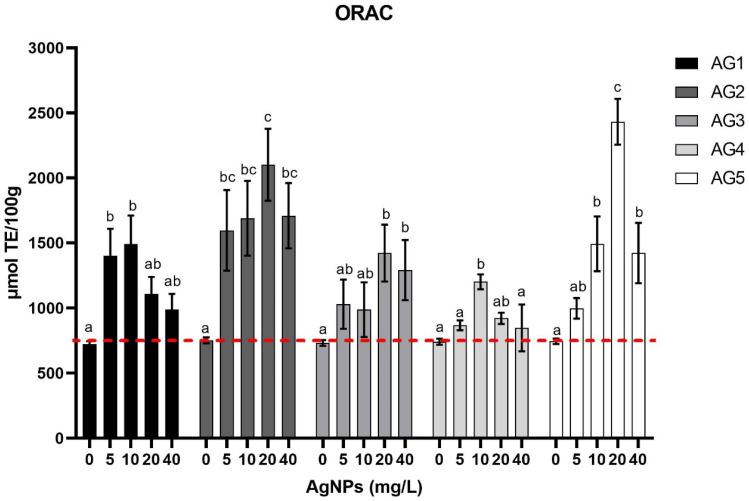
Effect of 3-min immersion in five types of silver nanoparticles (AgNPs) of carrots stored at 25 °C for 24 h. Samples were stimulated with 0, 5, 10, 20, or 40 mg/L of AgNPs, and antioxidant capacities (ORAC) were determined and quantified in Trolox equivalents (TE). Data represent the means of five replicates and their standard errors (*p* < 0.05). The red line indicates the antioxidant capacity obtained in the treatment without nanoparticles.

**Figure 2 nanomaterials-11-03148-f002:**
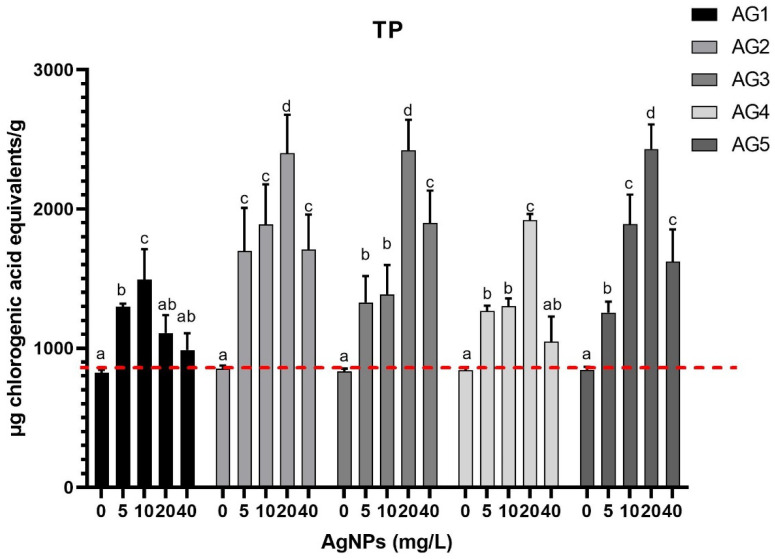
Effect of 30-min immersion in five types of AgNPs of carrots stored at 25 °C for 24 h. Samples were stimulated with 0, 5, 10, 20, or 40 mg/L of AgNPs and total phenolic contents (TP) were determined by Folin-Ciocalteu method. Data represent the means of five replicates and their standard errors (*p* < 0.05). The red line indicates the total phenolic content obtained in the treatment without nanoparticles.

**Figure 3 nanomaterials-11-03148-f003:**
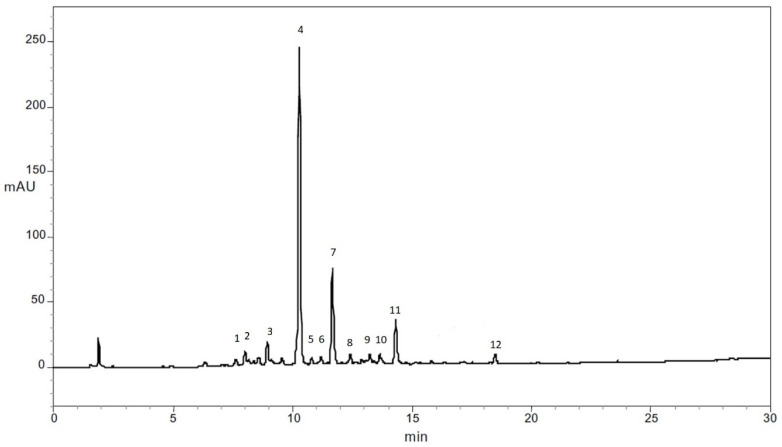
Typical HPLC chromatogram of phenolic compounds in carrots obtained at 280 nm. Peak identification: (1) chlorogenic acid; (2) 3-*O*-caffeoylquinic acid; (3) 5′-caffeoylquinic acid; (4) ferulic acid; (5) caffeic acid; (6) cis-5′-caffeoylquinic acid; (7) 4′p-coumaroylquinic acid; (8) 3-*O*-feruloylquinic acid; (9) 5-*O*-Feruloylquinic acid; (10) caffeic acid derivative; (11) 3′4′-dicafferoylquinic acid; (12) 3′5′-dicafferoylquinic acid.

**Figure 4 nanomaterials-11-03148-f004:**
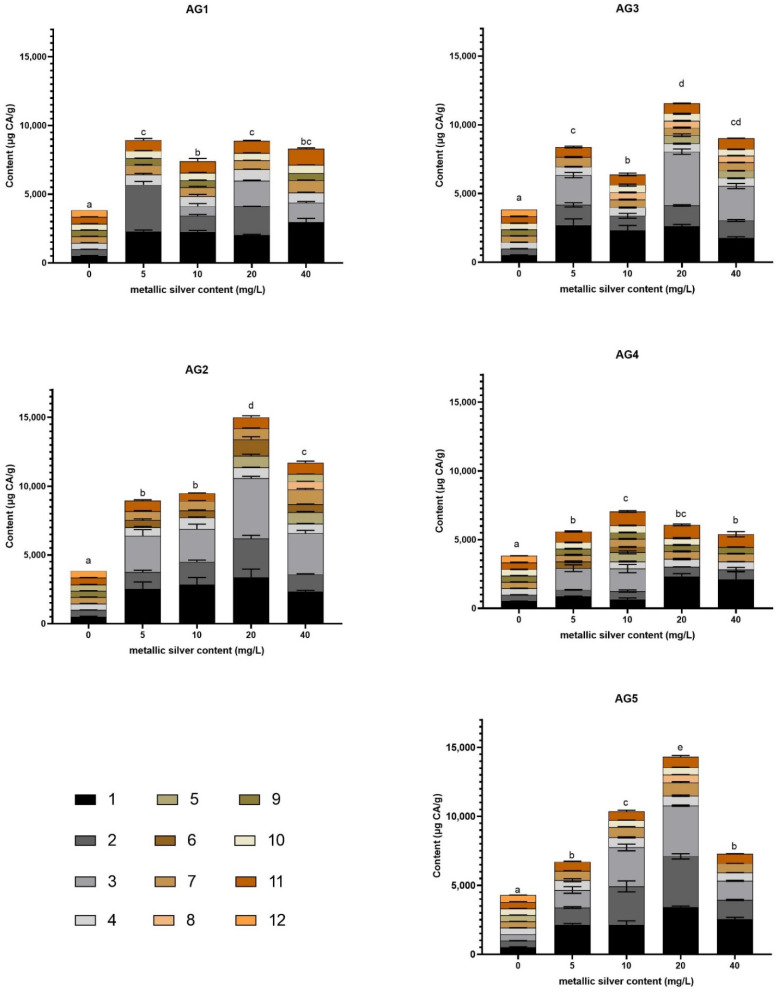
HPLC quantification of phenolic compounds in carrots stimulated with AG 1, AG2, AG3, AG4, and AG5. The quantified compounds were: (1) chlorogenic acid; (2) 3-*O*-caffeoylquinic acid; (3)5′-caffeoylquinic acid; (4) ferulic acid; (5) caffeic acid; (6) cis-5′-caffeoylquinic acid; (7) 4′p-coumaroylquinic acid; (8) 3-*O*-feruloylquinic acid; (9) 5-*O*-feruloylquinic acid; (10) caffeic acid derivative; (11) 3′4′-dicafferoylquinic acid; (12) 3′5′-dicafferoylquinic acid. Phenolic compounds were quantified in chlorogenic acid equivalents (CA) since it was the most abundant compound.

**Figure 5 nanomaterials-11-03148-f005:**
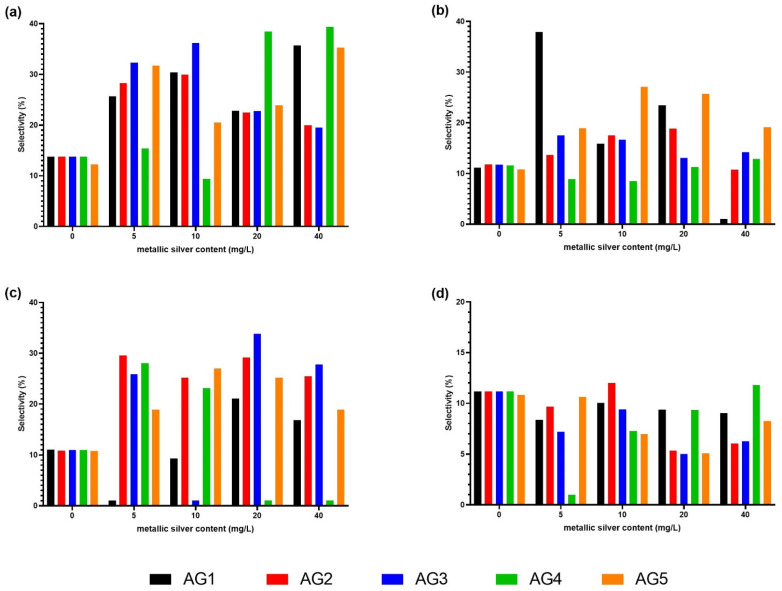
Selectivity of the most overproduced phenolic compounds with pharmaceutical interest induced by silver nanoparticles Argovit^®^. The quantified compounds were: (**a**) chlorogenic acid; (**b**) 3-*O*-caffeoylquinic acid; (**c**) 5′-caffeoylquinic acid; (**d**) ferulic acid.

**Table 1 nanomaterials-11-03148-t001:** Mass spectral characteristics of identified phenolic from carrot samples.

ID	Compound	[M-H] (m/z)	Fragments
1	Chlorogenic Acid	353	M_S2_ [353]: 345, 255, 147
2	3-*O*-caffeoylquinic acid	353	M_S2_ [353]: 135, 179, 191 M_S3_ [353→191]
3	5′-caffeoylquinic acid	353	M_S2_ [353]: 179, 191 M_S3_ [353→191]
4	Ferulic acid	193	
5	Caffeic acid	179.9	
6	Cis-5′-caffeoylquinic acid	353	M_S2_ [353]: 135, 179, 191
7	4′p-Coumaroylquinic acid	337	M_S2_ [337]: 191
8	3-*O*-Feruloylquinic acid	367	M_S2_ [367]: 173, 193 M_S3_ [367→173]
9	5-*O*-Feruloylquinic acid	367	M_S2_ [367]: 191 M_S3_ [367→191]
10	Caffeic acid derivative	367	M_S2_ [367]: 135, 179, 191 M_S3_ [367→179]
11	3′4′-Dicafferoylquinic acid	527	M_S2_ [527]: 203, 365 M_S3_ [527→365]
12	3′5′-Dicafferoylquinic acid	515	M_S2_ [515]: 353 M_S3_ [515→353]

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
