# Peer review of "Treatment with Argovit® Silver Nanoparticles Induces Differentiated Postharvest Biosynthesis of Compounds with Pharmaceutical Interest in Carrot (Daucus carota L.)"

_nanomaterials, 2021, doi:10.3390/nano11113148_

Round 1
Reviewer 1 Report
Manuscript can be accepted
Author Response
Dear Reviewer 1:
We received your comment and we appreciate it. I am submitting online the manuscript entitled "Treatment with Argovit® silver nanoparticles induce differentiated postharvest biosynthesis of compounds in carrot (Daucus carota L) with pharmaceutical interest" by Santoscoy-Berber et al.
The response to each observation is in the attached document.
Thank you so much for your time
Sincerely,
Rocío Alejandra Chávez Santoscoy, Ph.D.
Assistant Research Professor, SNI I
School of Engineering and Science
Tecnológico de Monterrey,
Monterrey, México 64849
Reviewer 2 Report
The work by R.A. Chavez-Santoscoy and coworkers describes the effect of AgNP exposure on the production of antioxidant secondary metabolites by Dancus carota L. The Introduction is clear, concise and well written, while the Results section needs revision. It contains both typos and several incorrect sentences. It would be advisable to seek help from a native English-speaking scientist. In general, the topic of the paper is of interest to the readers of Nanomaterials, but the scientific soundness of the article is not enough to allow its publication in its present form. Besides, the novelty of the manuscript is quite poor, as the effect of AgNPs on the production of phenolic compounds has been previously reported for different substrates. For instance, see: Dariusz Kruszka, Aneta Sawikowska, Rajendran Kamalabai Selvakesavan, Paweł Krajewski, Piotr Kachlicki, Gregory Franklin, Silver nanoparticles affect phenolic and phytoalexin composition of Arabidopsis thaliana, Science of The Total Environment, Volume 716, 2020, 135361 DOI: 10.1016/j.scitotenv.2019.135361.
Major points are:
1) Results. The authors should clearly describe here the five types of AgNPs employed. What are the differences between them? The reader cannot follow the discussion if this is not explained.
2) Materials and Methods. The authors should explain further how they quantified the phenolic compounds. Paragraph 2.5 does not mention any calibration standard solutions. Did the authors perform quantification only on the basis of sample weight and absorption values? If so, this is not enough.
3) Lines 252-253 [...]"It is attributed to the fact that the profile of compounds produced in each treatment and each concentration were different (Fig. 4).": Besides the need of language revision, are the authors sure that their methanolic extract contains only phenolic compounds? Can the authors rule out the presence of other substances potentially able to act on the antioxidant capacity of the mixture?
Minor points:
Introduction
Lines 61-62, "Carrot ... abiotic stress.": please, add a reference in support to this statement.
Line 69, "Due to nanoparticles (NPs) are particles" ? Please, correct.
Lines 79-80 "Nevertheless, a deep understanding of the role of nanomaterials in plant physiology at the molecular level is still lacking." Such role is not described in the manuscript, then please shift the sentence to the end of the manuscript at a possible follow-up to the work.
Line 84, "PVP" : please, explain abbreviations when first encountered in the text.
Line 126, Materials and methods: " Oxygen Radical Absorbance Capacity (ORAC) assay".
Lines 174-175 "AgNPs Argovit were able to increase antioxidant capacity in a in a dose-dependent manner. It was observed the hormetic effect of AgNPs Argovit, because they have a bell-like behavior, characterized because the antioxidant capacity increase": Statement unclear. Please, rewrite it. English language of the second part of the sentence needs revision.
Discussion
Line 246, "The treatments with AG2 and AG5 were the silver nanoparticles..."?: statement unclear, please correct it.
Author Response
Dear Reviewer 2:
We received your comments, and we appreciate all your suggestions. We have responded in detail all the comments, and I am submitting online the manuscript entitled "Treatment with Argovit® silver nanoparticles induce differentiated postharvest biosynthesis of compounds in carrot (Daucus carota L) with pharmaceutical interest" by Santoscoy-Berber et al. for your consideration.
The response to each observation is in the attached document.
Thank you so much for your time.
Sincerely,
Rocío Alejandra Chávez Santoscoy, Ph.D.
School of Engineering and Science
Tecnológico de Monterrey,
Monterrey, México 64849

Reviewer 3 Report
Dear Authors,
The Manuscript ID: nanomaterials-1456865 is interesting and it could be published after a minor revision. I made myself some suggestions/corrections in the attached manuscript.
Authors must pay attention to the red and yellow highlighted words/sentences. Thus, the red highlighted words/sentences represent corrections & suggestions. The yellow highlighted words show that the division into syllables was not done correctly.
Briefly, I suggested the following:
- Moderate English changes required.
- In Introduction, the authors must highlight the novelty/originality of their work.
- The Latin words must be written italic (see Lines: 36; 72; 73; 344; 346; 356; 364; 370; 378; 379; 381; 384; 390; 396; 400; 407; 421).
- Line 27: Replace “nanoparticle” with “nanoparticles”.
- All abbreviations must be detailed when they first appear in the text (see Line: 31).
- Reformulate the sentences from the Lines: 31-33; 102-103; 175-176; 186; 206-207; 317).
- Line 82: Replace “project” with “study”.
- Line 101-102: Replace “slides” with “slices”.
- Replace “ml” with “mL” (see Lines: 111; 146).
- Line 122: Delete “and”.
- Line 158-159: Replace “dierences” with “differences”.
- Line 164: Replace “present” with “presents”.
- Line 165: Replace “for” with “four”.
- Line 174: Delete “in a”.
- Line 205: Delete “.”.
- Line 211: Replace “chromatograph” with “chromatogram”.
- Line 221: Give more details about the negative control treatment.
- Line 224: Specify the method of Quantification of phenolic compounds in carrots!
- Line 227: Replace “concentration” with “Concentration”.
- Line 230: In the section: "Materials and Methods", the authors must give the expression for selectivity calculation.
- Line 230: Replace “identifided” with “identified”.
- Line 241: How did the authors determine the concentration of AgNPs?
- Line 315: Replace “if” with “of”.
- Did the authors study how much AgNPs was taken up by carrots? Did they quantify the cellular uptake of AgNPs by carrot cells? At certain concentrations, the presence of AgNPs could be toxic to people who eat carrots treated with AgNPs. Please, give more details regarding these issues.
- In the section “References”, please insert the names of all authors (See reference 5).

Author Response
Dear Reviewer 3:
We received your comments, and we appreciate all your suggestions. We have responded in detail all the comments, and I am submitting online the manuscript entitled "Treatment with Argovit® silver nanoparticles induce differentiated postharvest biosynthesis of compounds in carrot (Daucus carota L) with pharmaceutical interest" by Santoscoy-Berber et al. for your consideration.
The response to each observation is in the attached document.
Thank you so much for your time
Sincerely,
Rocío Alejandra Chávez Santoscoy, Ph.D.
Assistant Research Professor, SNI I
School of Engineering and Science
Tecnológico de Monterrey,
Monterrey, México 64849

Round 2
Reviewer 2 Report
The revised version of the manuscript is suitable for publication on Nanomaterials.